# Holistic Design of Experiments Using an Integrated Process Model

**Thomas Oberleitner [1]** , **Thomas Zahel [2]** , **Barbara Pretzner [2]** and **Christoph Herwig [2,3,]***

1    Competence Center CHASE GmbH, Ghegastraße 3, Top 3.2, 1030 Vienna, Austria
2    Körber Pharma Austria GmbH, PAS-X Savvy, Mariahilferstraße 88A/1/9, 1070 Vienna, Austria
3    Research Area Biochemical Engineering, Vienna University of Technology, Gumpendorferstrasse 1a, 1060 Vienna, Austria
*    Correspondence: christoph.herwig@tuwien.ac.at

**Abstract:** Statistical experimental designs such as factorial, optimal, or definitive screening designs represent the state of the art in biopharmaceutical process characterization. However, such methods alone do not leverage the fact that processes operate as a mutual interplay of multiple steps. Instead, they aim to investigate only one process step at a time. Here, we want to develop a new experimental design method that seeks to gain information about final product quality, placing the right type of run at the right unit operation. This is done by minimizing the simulated out-of-specification rate of an integrated process model comprised of a chain of regression models that map process parameters to critical quality attributes for each unit operation. Unit operation models are connected by passing their response to the next unit operation model as a load parameter, as is done in real-world manufacturing processes. The proposed holistic DoE (hDoE) method is benchmarked against standard process characterization approaches in a set of in silico simulation studies where data are generated by different ground truth processes to illustrate the validity over a range of scenarios. Results show that the hDoE approach leads to a >50% decrease in experiments, even for simple cases, and, at the same time, achieves the main goal of process development, validation, and manufacturing to consistently deliver product quality.

**Keywords:** design of experiments; holistic experimental design; integrated process model; optimal designs; process characterization; biopharmaceutical process validation

## 1. Introduction

The goal of process characterization in biopharmaceutical development is to establish scientific evidence that a process is able to consistently deliver quality products. An important part of this procedure is to determine the effect of process parameters (PP) on critical quality attributes (CQA [1,2]). Design of experiments (DoE) is a well-established tool to design experimental runs that yield such information and is oftentimes followed by data analysis and inference based on regression models [3,4]. DoE variants such as factorial or optimal designs facilitate the detection of effects by minimizing or eliminating correlation, and they are comprised of all possible combinations of effect levels or a subset thereof [5]. These experiments are then conducted in small-scale models for each unit operation (UO), and results can be used to create mathematical models that quantify the impact of effects.

Of particular interest when defining a control strategy is the range in which PPs can safely operate while keeping CQA concentrations within acceptable boundaries. Those proven acceptable ranges (PAR) are part of the control strategy a manufacturer might submit to a regulatory authority [6]. One way of defining a parameter's PAR is illustrated in Figure 1. Using this approach, the PAR is affected by two components: first, the model prediction, including the statistical intervals and second, the acceptance limits, as the intersection between the two defines the PAR's range. Another option could be to alter the setpoint condition of other PPs (see Section 4.2). This leads to a change in the

univariate prediction plot by shifting the prediction vertically (orange line in Figure 1). However, optimization by changing setpoint conditions is usually not the focus of process characterization. The PAR is required to allow for sufficient process and operator variability while being conservative enough to keep CQAs within acceptance limits. If the PAR is too narrow for adequate operability, one can either try to reduce model uncertainty by investing DoE runs or change acceptance limits by performing spiking runs.

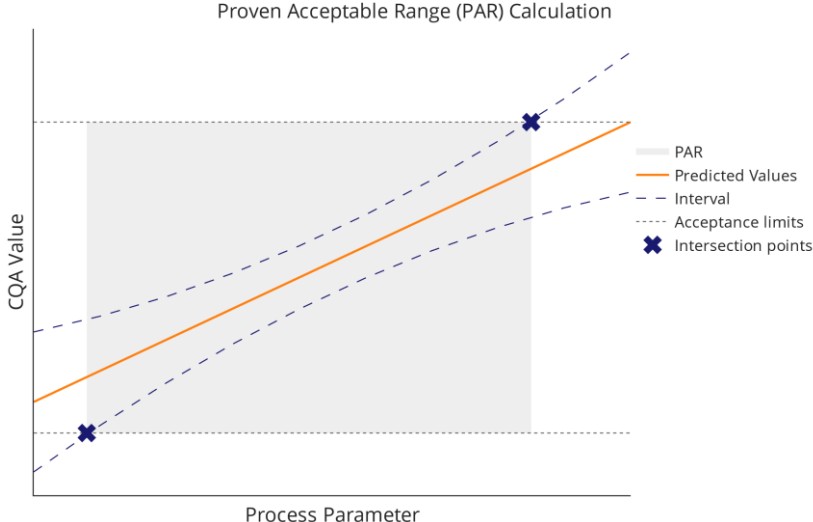

**Figure 1.** An example for how the PAR of a process parameter can be calculated. The predicted mean of the CQA as a function of the PP is shown in orange and the statistical interval around these predictions is illustrated as dashed lines. Lower and upper PAR boundaries can be defined by the intersection points of a statistical interval and the CQA acceptance criteria, marked as x.

### 1.1. Option A: Improving Model Estimates via DoE

To illustrate the effect of DoE runs on model uncertainty, consider the formula for calculating tolerance intervals for a normally distributed population [7]:

$$\hat{y} \pm z_{\frac{1+\psi}{2}} \sqrt{\frac{\nu\left(1 + \left(\frac{1}{N}\right)\right)}{\chi^2_{\alpha,\nu}}} \sigma \tag{1}$$

$\hat{y}$ is the mean prediction of the model, $1 - \psi$ the nominal proportion of the population covered by the interval, and $\alpha$ the confidence level. Disregarding the critical value for the normal distribution $z_{\frac{1+\psi}{2}}$ and the standard deviation $\sigma$, the dominant factor in this formula is the square root term, which includes the lower $\alpha$ quantile of the $\chi^2$ distribution in its denominator and converges toward one as $N$ increases. The residual degrees of freedom in a regression model are calculated as $\nu = N - p$, where $p$ is the number of model parameters. Note that this is a simplified version of the tolerance interval, and other methods might be used in a regression setting [8–10]. The graph of the square root term is shown in Figure 2 over a range of possible values of $N$.

The figure illustrates the strong decrease in this factor for the first values of $N$ before the curve starts to flatten. While other measures of model quality, e.g., the standard deviation or parameter covariance are contributing factors in a regression setting, this effect is representative of the behavior of an interval as the number of observations increases. For the experimental effort invested in improving parameter estimates and model quality this means that at some point no large improvements can be achieved in the interval width and, in turn, the PAR. Then, tackling the second decisive element, the intermediate acceptance criteria, might be more rewarding.

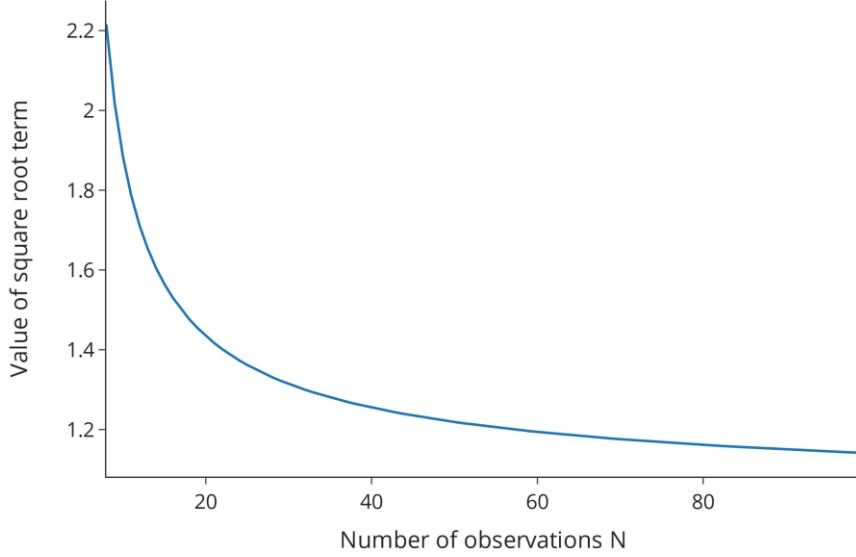

**Figure 2.** Value of the term $\sqrt{\frac{\nu\left(1+\left(\frac{1}{N}\right)\right)}{\chi^2_{\alpha,\nu}}}$ over a range of values for *N*.

### 1.2. Option B: Improving Acceptance Limits via Spiking Studies

In typical biopharmaceutical process development and characterization, unit operations are studied individually, and acceptance criteria need to be defined for each UO. Those intermediate acceptance criteria (iAC) are the second component affecting the PAR calculation, as shown in Figure 1. A frequently followed but flawed approach to setting iACs is to calculate this range using three standard deviations (SD) of manufacturing scale runs [11–13]. A much more scientifically sound method has recently been published where the only requirement is to have drug substance/product specification. In this approach, an IPM is used, and iACs for all UOs can be calculated inversely, starting from the specifications [11]. The same approach is shown here in an illustrative manner: one can correlate the inputs/loads and outputs/pools of each unit operation, as shown in Figure 3. If the slope of this correlation equals one, all of the load will be found in the pool, which is not desirable in a downstream UO. If the slope equals zero, the same (low) pool value will be achieved regardless of the load values, which is an ideal and robust scenario of a downstream UO. We can now calculate the iAC by backpropagating the iAC of the next UO (starting with the DS specifications) through those models. By iteratively applying this technique, the iACs for the entire process can be calculated. As all models are data-based, conservative extrapolation needs to be taken into account when making predictions outside the observed training space of the explanatory variables (here, the input material of each model). As discussed in [14], for impurities, we assume that every additional amount of loaded impurity beyond the observed range will be directly propagated into the output/pool of the UO, which can be mathematically seen as a piecewise regression model with a slope of one (see Figure 3).

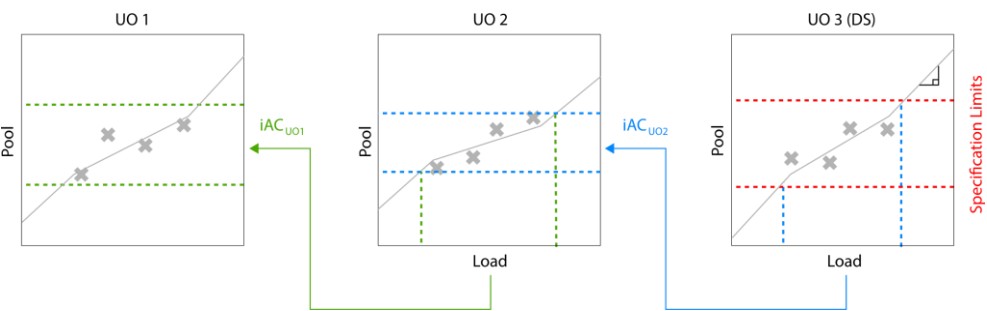

**Figure 3.** Inverse iAC calculation, beginning with drug specifications at the last UO.

Figure 4 illustrates the effect of adding a spiking run that shows successful clearance. The data point is added at UO 3 and the observed trend will be extrapolated, leading to an increase in iAC in UO 2. Since the iAC of UO 2 is used to calculate the iAC of UO 1, its iAC will be increased as well, etc. Hence, introducing a spiking run at one UO will potentially lead to increase in iACs of all previous UOs. Of course, the addition of spiking runs at extreme levels will be limited by the clearance capacity of the downstream UO. Note that Figures 3 and 4 show a simplified version for calculating iACs and that the actual method can involve more advanced statistical methods such as tolerance intervals and Monte Carlo sampling (for details, see [11]). Furthermore, while a linear correlation is assumed here, any mathematical model can be used to describe the dependency between input and output of individual UOs.

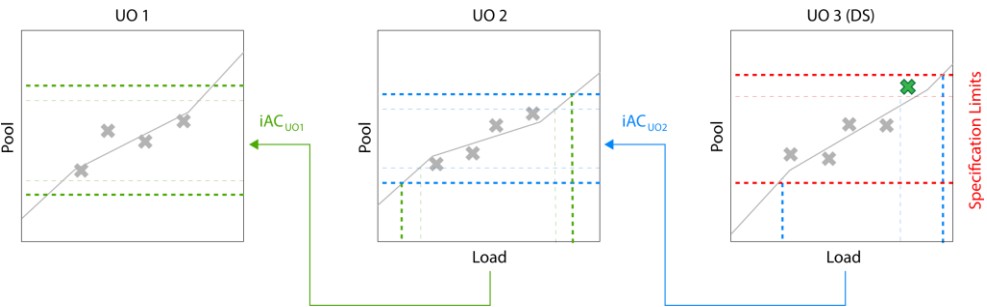

**Figure 4.** The effect of adding an additional data point on the iAC calculation.

Using spiking runs is not a novelty and commonly used in the industry [15,16]. However, results are usually only reported in documents and in our experience do not find entrance into mathematical modeling that also accounts for the uncertainty around individual experiments. Marschall et al. describes how any manufacturing or small-scale data can be used to calculate iACs and how spiking runs are included in that procedure [11].

At this point, we have demonstrated that both the addition of DoE runs to decrease model uncertainty as well as the addition of spiking runs to increase iACs can help to gain process understanding, which helps to increase PARs and facilitates a more flexible control strategy. However, it remains unclear which combination of DoE or spiking runs would give the maximum gain in PAR. Therefore, we want to:

- Develop a recommender system, called the holistic design of experiments (hDoE), that suggests the optimal runs (DoE or spiking) at specific UOs that lead to the fastest increase in process understanding. Here we define process understanding as the accuracy and precision of the (unknown) true relation between all PPs and CQAs, as well as the input/output relation of individual UOs. We describe this method in Section 3.1;
- Demonstrate that using such a recommender system can lead to a significant reduction in the required number of total runs of a process characterization study (PCS) using state-of-the-art workflows. We verify this in a set of simulation studies presented in Section 3.2;
- Identify an accelerated workflow for PCS using hDoE that can be applied in practice; see Section 4.1.

## 2. Materials and Methods

### 2.1. Optimal Designs

Factorial or fractional factorial designs are generally considered the ideal approach to creating experimental designs that yield the most information about how process parameters affect the response. An exhaustive account of such designs can be found in [5]. In practice, however, they are not universally applicable as they often require many runs and cannot incorporate existing data. Optimal designs constitute a more flexible alternative [17–20]. The number of runs required is not a consequence of the chosen type of

design, e.g., $2^k$, for a factorial design with two levels and $k$ parameters but can be chosen more flexibly. Furthermore, optimal designs can be used to augment an existing set of runs. Both of those properties are important for their application in hDoE, as the procedure starts off with a minimal set of experiments far smaller than a full factorial design, which gets augmented in each experiment/evaluation cycle. Based on the working set of already performed experiments, new runs based on optimal designs are recommended. Optimal designs optimize specific properties of the design matrix $X$. For example, D-optimal designs minimize the variance of parameter estimates in a model. As $Var(\hat{\beta}) = \sigma^2(X^T X)^{-1}$, this is equivalent to maximizing $|X^T X|$, the determinant of the squared design matrix and one can see that this is maximized when the columns of $X$ are orthogonal. However, in contrast to factorial designs, strict orthogonality is not required and one consequence of that is that the number of runs in $X$ can be set freely depending on the use case. The rows in $X$ are then chosen by exchange algorithms from a candidate set of runs, which generally consists of all possible level combinations for the main effects defined in the model. We used augmented D-optimal designs to generate the experiments recommended by hDoE.

### 2.2. Integrated Process Model

An integrated or holistic process model is an in-silico representation of a manufacturing process comprised of multiple steps or unit operations. While there are many approaches to constructing process models (see [21] for an overview), here we consider the IPM as an empirical ensemble model implemented as a sequence of UO models that enable predictions of different CQAs as a function of process parameters. To simulate CQA concentrations, the predicted values are passed on to the next UO as a process parameter in a Monte Carlo simulation that randomly draws parameter values [22]. This is done over the entire chain of UOs in the process, from upstream operations to the final drug substance. For a comprehensive description of the method, please consult [14].

An important aspect of this approach is extrapolation. Because parameter values are drawn randomly from their corresponding distribution, CQA predictions of one UO regularly exceed the range of values observed in the training data of the predecessor UO. This is a problem for the conservative prediction of CQA concentrations, as UOs are modeled as data-driven regression models. Such models are generally only valid within the range of the training data and extrapolation can lead to highly biased results [23].

### 2.3. Simulation Study

We investigated the performance of hDoE in a set of simulation studies illustrating different situations and processes. Results are compared to a state-of-the-art (SOTA) process characterization workflow that consists of conducting experiments based on optimal designs to investigate the impact of PPs on CQAs per UO. For the reference method, a D-optimal design was chosen with 6, 12, and 23 runs per UO, which leads to 24, 48, or 92 runs overall in 4 UOs.

Each process in this simulation study consists of a sequence of unit operations represented by ground truth equations that map PPs to a CQA. The first UO can be interpreted as the fermentation step, followed by three downstream UOs. The equations that describe these UOs satisfy IPM conditions by adhering to the heredity principle [24] and having linear load dependencies. We then try to find effect coefficients in the presence of added noise, employing both hDoE and the SOTA method that uses a predefined number of runs per UO, as described above. For hDoE, we start with a minimal design of 6 runs per UO (total 24 runs for 4 UOs) and add an additional 30 runs, chosen by the recommender system. Note that for the simulation study results reported here, only a single run was recommended per cycle, though results are similar for larger sets of run recommendations. To calculate OOS rates, an upper drug substance specification for the output/pool of UO 4 was set as three standard deviations above the mean of the ground truth process. Hence, when adding an infinite number of runs, a minimum OOS rate of 0.00135 ([1–0.9973]/2) can be achieved. As the OOS rate simulated by the IPM is based on drawing random

values from a PP distribution, mean and variance must be specified. In this normalized setting, each PP's setpoint, i.e., the mean of the distribution, was chosen to be zero and the variance was set to be the same as that of the observed ground truth data that were used to derive specification limits. This was kept constant over all simulated hDoE steps to avoid a misleading optimization trajectory that improves OOS rates by simply reducing PP variances without increasing process knowledge. Each simulation scenario was repeated 100 times with different random seeds. A summary of simulation parameters is provided in Table 1.

**Table 1.** Parameters for the simulation study.

| Parameter | Value |
|---|---|
| Number of UOs | 4 |
| Parameters per UO | 5 |
| CQA type | Impurity |
| hDoE start runs | 6 |
| Noise/std ratio for residual error | 0.5 (0.9 in study D) |
| Number of runs recommended per cycle | 1 |
| Variable selection method | Bi-directional stepwise |
| *p*-value threshold for including effect | 0.25 |
| *p*-values threshold for excluding effect | 0.05 |

The number of hDoE steps, repetitions of the simulation studies and the bi-directional stepwise variable selection method [25] for (re)fitting IPM models were chosen as a compromise between accuracy of results and simulation runtime.

For demonstration purposes but without loss of generality, we employ all simulation studies only for one CQA of the product. However, the methodology is not limited to the number of investigated CQAs. In practice, one would focus on the CQA, which shows the highest OOS rate.

### 2.3.1. Study A: Baseline

This simulation study represents a typical biopharmaceutical process with some quadratic and interaction effects and coefficients commonly found in characterization studies. In our experience, approximately 20–40% of all possible effects are practically significant in a model. A total of 5 factors lead to 20 effects (main, 2-factor interaction, and quadratic). In the ground truth, we have chosen 4–6 active effects, which equals 20–30% of all possible effects and is, therefore, within the expectation of a representative biopharmaceutical process. Moreover, we assume linear load dependencies and heredity between main and higher-order effects, which is also representative of the authors' experience. The ground truth equations for this scenario are as follows:

$$y_{UO1} = 8.0 + 0.7\, x_1 + 0.6\, x_2 + 0.5\, x_3 - 0.4\, x_4 + 0.9\, x_2^2 + 0.8\, x_3^2 \tag{2}$$

$$y_{UO2} = 3.4 + 0.5\, y_{UO1} + 0.5\, x_2 + 0.3\, x_3 + 0.5\, x_5 + 0.7\, x_2 x_3 + 0.4\, x_2 x_5 \tag{3}$$

$$y_{UO3} = 3.0 + 0.3\, y_{UO2} + 0.4\, x_1 - 0.3\, x_2 + 0.2\, x_4 - 0.2\, x_5 + 0.3\, x_1^2 - 0.7\, x_5^2 \tag{4}$$

$$y_{UO4} = 2.8 + 0.2\, y_{UO3} + 0.1\, x_1 + 0.2\, x_3 + 0.2\, x_5 + 0.3\, x_1 x_3 \tag{5}$$

### 2.3.2. Study B: Load Effect Set to One

As hDoE leverages the UOs dependency on the load, we investigated its behavior when the load coefficient is set to one, and its values are passed directly to the output of UO 3, provided other PPs are at setpoint. This mimics the situation where a full propagation of the CQA through this UO is expected, and no clearance takes place. This is, of course, not

the desired behavior of a downstream unit operation. In this setting, the load coefficient in UO 3 was set to one:

$$y_{UO3} = 3.0 + 1.0\ \mathbf{y_{UO2}} + 0.4\ x_1 - 0.3\ x_2 + 0.2\ x_4 - 0.2\ x_5 + 0.3\ x_1^2 - 0.7\ x_5^2 \tag{6}$$

2.3.3. Study C: All Load Effects Set to One

Here we set the load effects of all UOs to one. Every UOs output is directly propagated to the next UO, which means that there is no information about the load that could be detected by spiking runs. This a very untypical scenario as usually, we expect some clearance activity of the downstream UOs (UO 2–4). The purpose of this simulation is to show that, in the worst case, hDoE performs similarly to standard approaches using a predefined set of runs.

$$y_{UO2} = 3.4 + 1.0\ \mathbf{y_{UO1}} + 0.5\ x_2 + 0.3\ x_3 + 0.5\ x_5 + 0.7\ x_2x_3 + 0.4\ x_2x_5 \tag{7}$$

$$y_{UO3} = 3.0 + 1.0\ \mathbf{y_{UO2}} + 0.4\ x_1 - 0.3\ x_2 + 0.2\ x_4 - 0.2\ x_5 + 0.3\ x_1^2 - 0.7\ x_5^2 \tag{8}$$

$$y_{UO4} = 2.8 + 1.0\ \mathbf{y_{UO3}} + 0.1\ x_1 + 0.2\ x_3 + 0.2\ x_5 + 0.3\ x_1x_3 \tag{9}$$

2.3.4. Study D: Disabled Probability-Ratio-Threshold

To highlight the importance of the probability-ratio-threshold (PRT) decision scheme, a method borrowed from the Metropolis–Hastings algorithm that encourages the detection of new effects (described in Section 3.1), we repeat the baseline study without PRT. Here, the decision logic simply recommends the type of run that leads to the largest reduction in OOS. As PRT is most effective in situations where no clear decision can be made due to high residual error, the error/standard deviation ratio in the ground truth was increased from 0.5 to 0.9.

## 3. Results

### 3.1. Holistic Design of Experiments

In this contribution, we propose a new tool for process development and characterization: holistic design of experiments (hDoE), an iterative approach to experimental design and evaluation that minimizes the number of runs invested while maximizing the overall process understanding, as defined in the introduction. As all UOs of the process contribute to the generation of DS material, we can boil down process understanding to how well the true distribution of DS product quality is known. To express this in a single measure that can be used for optimization, we chose the out-of-specification (OOS) rate based on known DS specifications.

As described in Section 2.2, the IPM connects UOs by passing the output of a UO to the next one as a load parameter. While the output might be affected by different parameters and interactions, the univariate relationship between load and output is assumed to be linear inside the range of observed load values in the training data. However, in the Monte Carlo simulation of a CQA distribution, simulated load values might exceed this observed range, in which case the CQA value is handled conservatively to avoid predictions that are too optimistic (see piecewise load model in Taylor et al. [14]). This means that the simulated CQA distribution in drug substances is highly dependent on the range of load parameter values in the training data. Small variations in the training data will lead to a broadened CQA distribution, accounting for the uncertainty due to extrapolation. However, in many cases, missing information about load parameters can be supplemented by performing spiking studies where PPs are kept at setpoint, and only the load is varied. This is in contrast to classic DoE studies, which examine specific combinations of factors and disregard the load.

Spiking and DoE runs are the two different types of the experiment proposed by hDoE in this contribution. Assuming model parameters do not change from one iteration to the next, DoE runs are expected to improve general process parameter estimates, while

spiking studies improve estimates of the model's load coefficient and reduce extrapolation in the IPM simulation. However, the assumption of unchanging model parameters is regularly violated when variable selection on newly acquired data results in a new model. As a consequence, the simulated OOS probability is not guaranteed to be improved in every iteration.

hDoE starts out by roughly characterizing the process with a minimal D-optimal design per UO that facilitates fitting the initial regression models. Based on the information acquired in this first step, a set of runs is proposed. This set consists of DoE and spiking runs and includes the target UOs in which to perform them. The runs are chosen by how much they would reduce the OOS probability calculated by the IPM. After conducting the proposed experiments, the IPM data are supplemented with new information, and a variable selection step updates models where appropriate.

As OOS predictions and therefore run suggestions are based on models found in previous steps, the process is biased toward already detected effects. Additionally, the decision rule is susceptible to noise, especially in early steps, and might consider the value of adding spiking or DoE runs equivalent. Of course, this is generally not true, as only DoE experiments enable the detection of new or interaction effects. To mitigate this bias and encourage the detection of new effects, we employ a technique based on a decision scheme used in the Metropolis–Hastings algorithm [26]. Let $X$ be the design matrix of the data already incorporated into the IPM and $x_{DOE}$ and $x_{spiking}$ be new DoE and spiking samples, respectively, chosen from a set of sample candidates that result in the lowest OOS probability. We then calculate the ratio of those probabilities $\alpha = P(x_{DOE}|X)/P(x_{spiking}|X)$, draw a number from a uniform distribution, $u \in [0,\ 1]$, and only suggest a spiking run when $u \geq \alpha$. This means that DoE runs are always recommended if $P(x_{DOE}|X) \geq P(x_{spiking}|X)$. Spiking runs, however, are only suggested when the improvement in OOS probability considerably exceeds that of a DoE run. In the following, we term this the probability-ratio-threshold (PRT) approach. In our evaluation of the algorithm, PRT generally circumvents the problem of selecting spiking runs unnecessarily or overlooking effects (see Section 3.2). Figure 5 illustrates the individual steps and decision processes involved in hDoE.

### 3.2. Simulation Results

3.2.1. Out-of-Specification Rates

Figure 6 shows that in most scenarios, a high process understanding (quantified as a low OOS rate) can be achieved with a much smaller number of experiments compared to the SOTA method that uses a fixed number of DoE runs (in this case, 24, 48 and 92). This effect is most pronounced in study A, where the mean OOS rate drops to ~2.5% after only six additional runs recommended by hDoE (30 total), whereas 92 D-optimal runs calculated beforehand result in an OOS rate of ~7.5% due to the lack of exploration of load-to-pool dependencies. At first glance, this might appear as an unfair comparison, but workflows applied in the industry generally do not incorporate spiking runs in a mathematical framework to achieve a specific goal, e.g., establishing a control strategy. The effect of an UOs dependency on the load can also be seen in the results of study B, where variation around the OOS rate is larger due to setting the load coefficient to one in UO 3 in the ground truth. This means that no additional information about the load can be acquired in this UO, and the advantage of systematically recommending spiking studies at this UO is softened. Study C represents the worst case for hDoE, where the load has no effect in any unit operation, resulting in OOS rates close to that of the reference method. The larger variation shown in study D is due to increased noise in the ground truth. This, of course, affects hDoE as well as the reference method. However, note that variation is drastically increased when PRT is disabled (purple, dotted lines) compared to the recommended procedure that uses it when deciding on runs (blue, dotted lines).

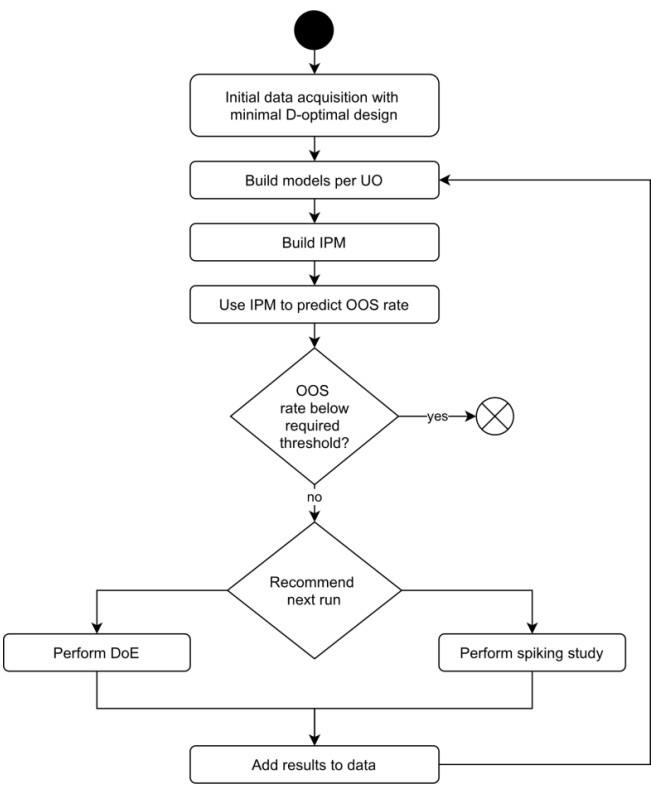

**Figure 5.** The hDoE recommender system that minimizes OOS by iteratively augmenting data and predicting optimal experiments.

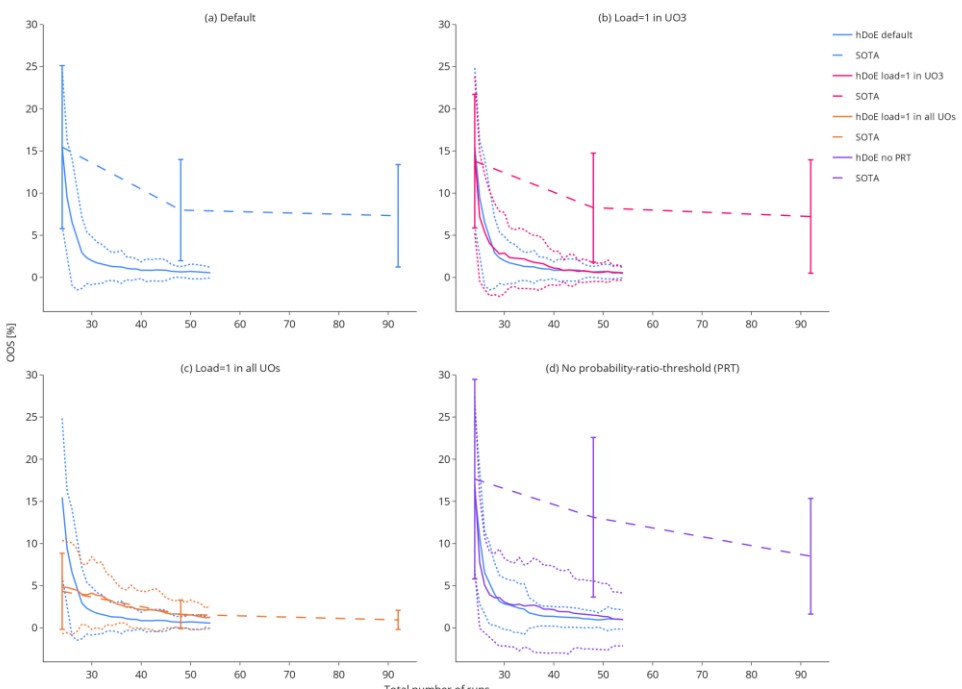

**Figure 6.** The decrease in OOS probability over the 30 hDoE steps for the different simulation scenarios. Each scenario was repeated 100 times. The solid line indicates the mean OOS probability, while the dotted lines show the standard deviation in the 100 repetitions of that step. For the reference method, OOS probabilities are plotted as dashed lines and were calculated at 24, 48, and 92 total runs.

### 3.2.2. Run Allocation

The drastic decrease in OOS probability over the number of experimental runs invested, shown in Figure 6, is achieved by performing spiking studies at the right UO. Generally, spiking runs are favored in the early steps of the procedure, as they eliminate extrapolation in the IPM (see Section 2.2) and therefore lead to the largest reduction in OOS early on. After this initial phase, larger OOS improvements can be achieved by improving parameter estimates, prompting hDoE to suggest more DoE runs. Figure 7 illustrates the allocation of different run types to the four unit operations of the simulation study. In the *y*-axis, the plots show the cumulative number of allocated runs over 100 repetitions of a simulation study, while the corresponding hDoE step can be seen in the *x*-axis. As the first UO (e.g., fermentation) is not affected by a load parameter, no spiking runs (dotted lines) are allocated. In the three simulation studies where the load influences a UO, spiking runs at UO 4 are recommended in the early steps and, in many of the 100 iterations, also in UO 3. This makes sense, as the OOS probability in drug substance, i.e., the last UO, is the main driver of the recommender system and its load coefficient directly affects the CQA distribution. In the absence of load effects, DoE runs are distributed approximately equally, while some spiking runs are accumulated due to noise (orange lines). The effect of increased noise and the absence of PRT are illustrated by the purple line. Much more spiking runs are invested, as the OOS simulation for the two types of experiment yields similar results, and choices are made more randomly.

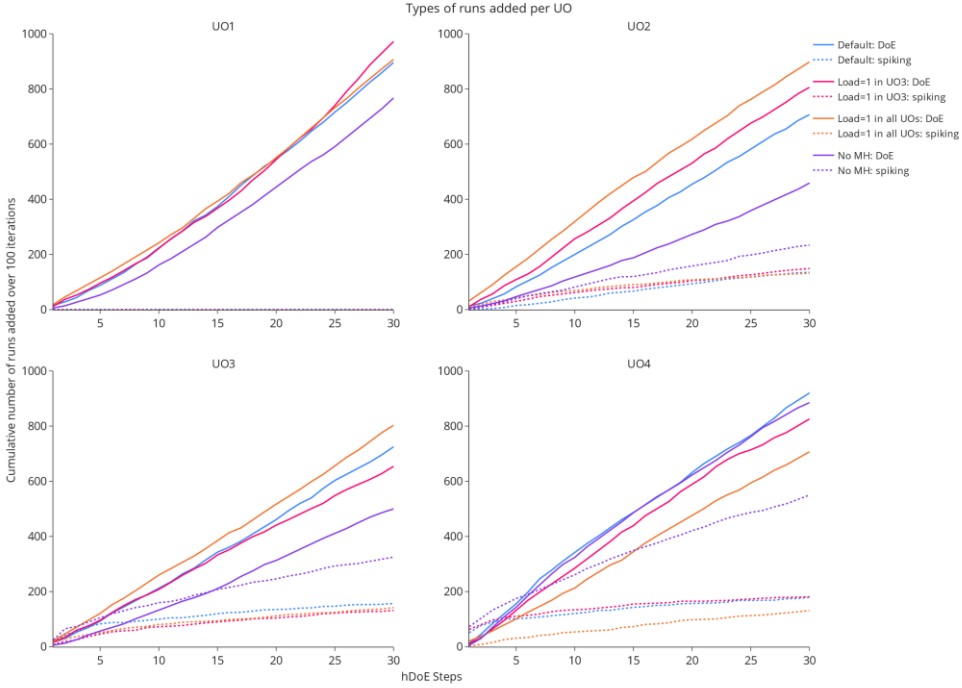

**Figure 7.** The cumulative allocation of either spiking or DoE runs in 100 repetitions of the simulation is shown over each step taken by the hDoE procedure (after the initial 24 runs). In total, each study distributes $100 \times 30 = 3000$ runs of any type to the four unit operations.

### 3.2.3. Parameter Estimates

A low OOS rate alone does not indicate correct models, as it does not account for aliasing effects in the parameters. Figure 8 shows the distributions of effects identified in the variable selection procedure at the last hDoE step (54 total runs invested). Each data point represents the effect size in a particular iteration of the simulation. We added an estimator of the residual variance in the form of the root mean squared error (RMSE). The mean estimated value (blue dashed line of each boxplot) converges to the ground truth (red solid line of each boxplot) for most parameters. Higher-order effect estimates in unit operations two and three are biased toward zero, i.e., they were not detected in variable

selection. Note that stepwise variable selection was used for performance reasons in this simulation, which is known to eliminate effects prematurely [27], and that some of this bias could be mitigated by using more modern approaches such as leaps and bounds or other exhaustive algorithms. However, as tolerance intervals are used in the estimation of model uncertainty [14], overlooking individual effects, which results in larger estimates of the RMSE, is accounted for correctly in the uncertainty interval. Of course, the quality of parameter estimates is a direct consequence of the number of DoE runs invested, which in turn is dependent on when the hDoE procedure is terminated.

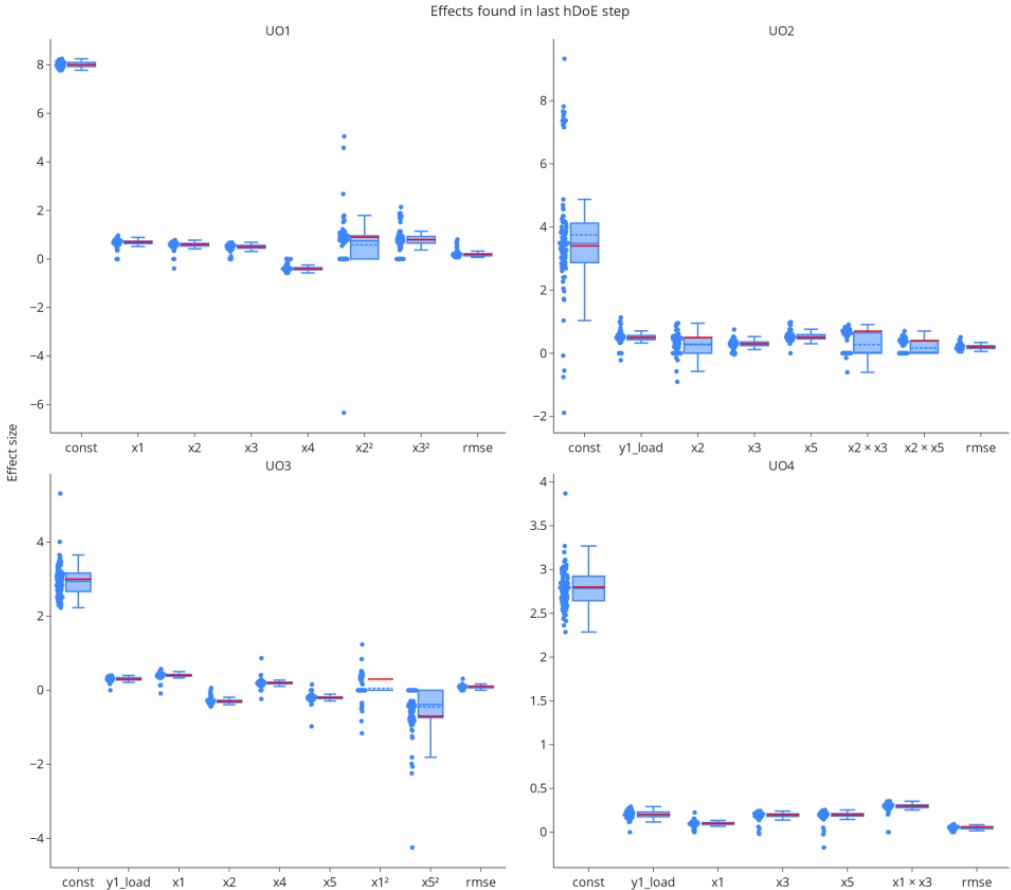

**Figure 8.** The distribution of effects found by stepwise variable selection at the last hDoE step. Each point represents the effect size in one of the 100 repetitions of the simulation study. The RMSE was added as an estimator of residual error.

Parameter estimates, especially for higher-order effects, are naturally not on par with those from a full D-optimal design with 23 runs per unit operation, see Figures 8 and 9. However, due to spiking studies invested in the characterization of the process, hDoE results in the improved estimation of load and intercept effects.

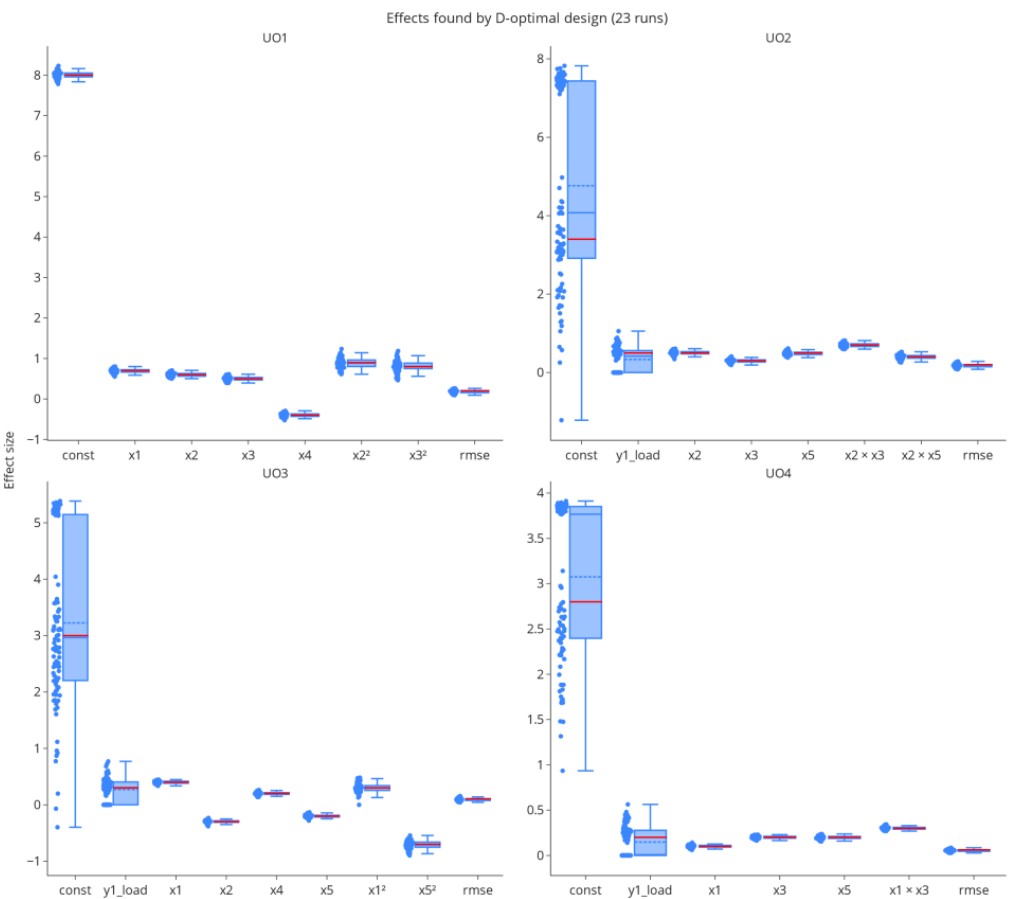

**Figure 9.** Parameter estimates for the full D-optimal design with 23 runs per UO. Variance in estimates is smaller than in hDoE. However, intercept and load effects are biased due to missing load data.

## 4. Discussion

### 4.1. hDoE in Process Characterization

Figure 10 shows how hDoE affects common steps in process characterization. The result of this procedure is a control strategy, of which PARs are an essential component. We describe in Section 1 the different courses of action when a PAR is too narrow to be part of an appropriate control strategy, steps that are also reflected by the hDoE recommender system. A third option that is currently not incorporated into the recommender system is to change the setpoint of other PPs that are active in the UO model. We present this approach as an outlook in Section 4.2. Finally, PAR ranges can be increased by reducing confidence/coverage levels of the statistical interval, although in most cases, this is not recommended and only mentioned here for the sake of completeness. We consider the workflow shown in Figure 10 as an extended version of the state-of-the-art workflow for process characterization (left column) that incorporates hDoE (right column, green boxes). This aligns with both the FDA and EMA guidelines for process characterization, as the former recommends DoE to increase process knowledge [3] and the latter impurity spiking challenges for downstream operations [28]. hDoE employed in the context of this extended workflow integrates both types of experiments and provides a systematic method for when and where to perform them.

### 4.2. Outlook: Changing PP Setpoints to Increase the PAR

An important aspect of the PAR calculation illustrated in Figure 4 is the univariate nature of this method. Only the screening range of the current PP is considered, while all other PPs are kept at their setpoint. As the UO model output, i.e., the CQA, is in most cases affected by multiple PPs, their setpoint can influence the offset of the univariate mean

prediction of the target PP (orange line in the figure) significantly. Consequently, a change in the setpoint of another PP can push the predicted CQA distribution inside/outside the acceptance limits and change the OOS rate. Similarly, interaction effects with other PPs can also influence the target PPs effect.

The optimization of PP setpoints is already available in some statistical software [29], and hDoE could be easily extended to include such recommendations based on their effect on the OOS rate of the process. While this is generally not the focus of process characterization studies, the FDA recommends optimization based on setpoint shifts in the continuous verification phase [3]. This would advance the applicability of the hDoE recommender system into the domain of process optimization.

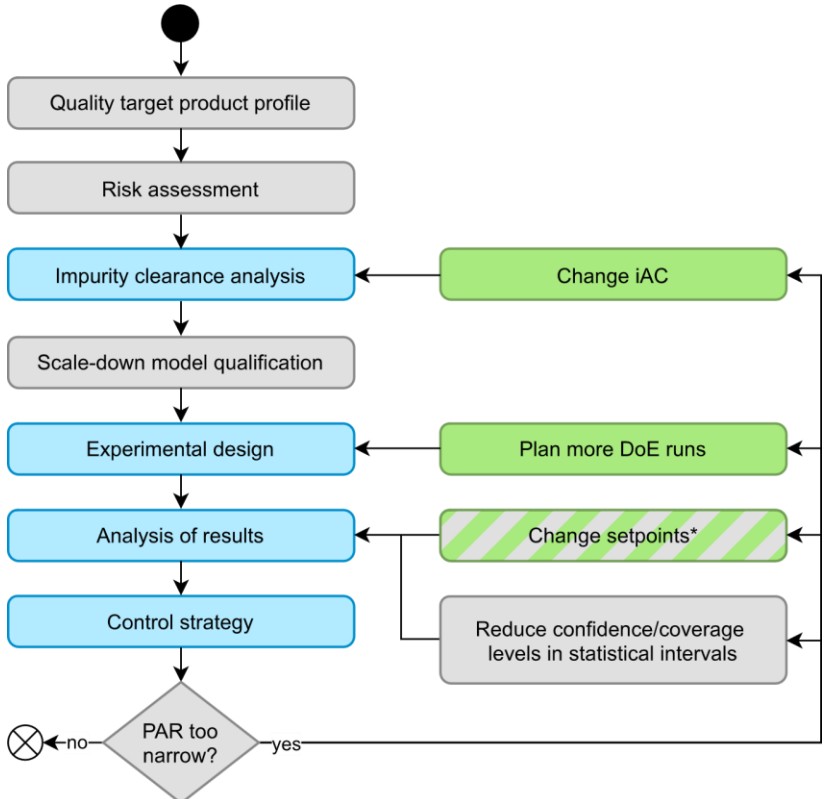

**Figure 10.** The left column represents common steps involved in process characterization, while the right column contains possible courses of action when no appropriate control strategy can be found. Green boxes are actions recommended by hDoE, and steps affected by hDoe are colored blue. * Changing PP setpoints could also be a viable, recommended action by future iterations of hDoE (see Section 4.2).

## 5. Conclusions

In this article, we introduced hDoE, an iterative tool for process development and characterization that facilitates a more effective way of gaining process understanding related to final product quality. This is essential, as it reduces experimental effort and time to market. To demonstrate the benefits of this approach quantitatively, we presented results from simulation studies where we chose the OOS rate as a measure of process understanding, which should be close to the OOS rate of the true (usually unknown) process. The benefit of this measure is that it includes both the mean and variability of the final product quality distribution as well as practically relevant limits (drug specifications). Other measures, such as the Kullback–Leibler divergence [30], also could have been used. However, we believe the OOS rate represents a more practically relevant measure and might be more tangible for process experts. We have demonstrated that hDoE leads to better overall process understanding with more than a 50% reduction in the number of

experiments performed for simple scenarios. The reduction of experimental costs can even be increased for specific cases. hDoE starts out with an initial, minimal set of D-optimal runs on which the first set of unit operation regression models is fitted. Used as a recommender tool, either DoE or spiking runs are added in an iterative fashion guided by improvements in the predicted OOS rate. As the process is biased toward effects already found, we are using a recommender scheme akin to the one used in the Metropolis–Hastings algorithm to promote the detection of unknown effects and to improve parameter estimates. Of course, the overall quality of effect estimates is influenced by the number of runs available to the algorithm. However, our simulation studies show that a compromise between effects detected and runs invested can be found using a relatively low number of hDoE runs.

hDoE leverages the link between UOs as modeled by the IPM and thereby improves OOS rates by strategically recommending spiking studies at specific process steps. It provides valuable information to biopharmaceutical manufacturers about which type of experiment to perform next, and in which UO, and can decrease the time and money invested in experimental design. Potentially increased parameter uncertainty due to a lower number of samples is accounted for in the IPM's OOS prediction using conservative sampling and estimation methods such as tolerance intervals. We believe that hDoE is a viable option for experimental design that yields robust estimates of process properties while providing better control of the resources invested. This will lead to a substantial reduction of development costs and time to market, ultimately leading to more affordable drugs.

**Author Contributions:** Conceptualization, T.Z. and B.P.; methodology, T.Z. and T.O.; software, T.O. and B.P.; validation, B.P. and T.O.; formal analysis, T.O.; investigation, T.O.; data curation, T.O.; writing—original draft preparation, T.O. and T.Z.; writing—review and editing, T.O., T.Z., B.P. and C.H.; visualization, T.O.; supervision, C.H. and T.Z.; project administration, T.Z. All authors have read and agreed to the published version of the manuscript.

**Funding:** This work was supported by the Austrian Research Promotion Agency (FFG) (grant number: 844608) and within the framework of the Competence Center CHASE GmbH, funded by the Austrian Research Promotion Agency (grant number 868615) as part of the COMET program (Competence Centers for Excellent Technologies) by BMVIT, BMDW, and the Federal Provinces of Upper Austria and Vienna. Open Access Funding by TU Wien.

**Institutional Review Board Statement:** Not applicable.

**Informed Consent Statement:** Not applicable.

**Data Availability Statement:** All data generated or analyzed during this study are included in this published article.

**Acknowledgments:** This work was conducted within the COMET Centre CHASE, funded within the COMET−Competence Centers for Excellent Technologies program by the BMK, the BMDW and the Federal Provinces of Upper Austria and Vienna. The COMET program is managed by the Austrian Research Promotion Agency (FFG). The authors acknowledge TU Wien Bibliothek for financial support through its Open Access Funding Program.

**Conflicts of Interest:** The authors declare no conflict of interest.

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
