# Peer review of "Holistic Design of Experiments Using an Integrated Process Model"

_bioengineering, doi:10.3390/bioengineering9110643_

Round 1

Reviewer 1 Report

The manuscript under the title Holistic Design of Experiments Using an Integrated Process Model , is proposing a holistic DoE (hDoE) statistical method in standard process characterization approaches in a set of in-silico simulation studies where data is generated by different ground truth processes to illustrate the validity over a range of scenarios.

The manuscript  demonstrated that both: a) the addition of DoE runs to decrease model uncertainty as well as the b) addition of spiking runs to increase iACs can help to gain process understanding, which helps to increase PAR (proven acceptable ranges) and facilitates a more flexible control strategy, but it still remains unclear which combination of DoE or spiking runs would give the maximum gain in PAR, and that was the topic of this manuscript. The authors wanted to develop a recommender system, called holistic design of experiments (hDoE), that suggests the optimal runs (DoE or spiking) at specific UOs that lead to the fastest increase in process understanding. Here we define process understanding as the accuracy and precision of the (unknown) true relation between all PPs (the effect of process parameters) and CQAs (critical quality attributes) as well as the input/output relation of individual UOs. The y explained what is important in Optimal Designs, what is and how to make and use Integrated Process Model and performed also Simulation Studies A, B, C and D.

At the end in the results they are proposing a new tool for process development and characterization: Holistic Design of Experiments (hDoE), an iterative approach to experimental design and evaluation that minimizes the number of runs invested while maximizing the overall process understanding as defined in the introduction.Spiking and DoE runs are the two different types of experiment proposed by hDoE in this contribution. they have tried to explain it on a simple way and to give a handbook to the users how to use this model (The hDoE recommender system that minimizes OOS by iteratively augmenting data and predicting optimal experiments) and what are the safe limits to reduce noise and to have acceptable and reliable results.

in the discussion part I can see now that in line 400 there is not a link on the section but writes error, so this part should be changed . Authors in the discussion part are explaining  different courses of action when a PAR is too narrow to be part of an appropriate control strategy - with the steps that are also reflected by the hDoE recommender system. A third option that is currently not incorporated into the recommender system is to change the setpoint of other PPs that are active in the UO model, and authors present this approach as an outlook in section 4.2. PAR ranges can be increased by reducing confidence/coverage levels of the statistical interval, although in most cases this is not recommended and only mentioned here for the sake of completeness.

authors consider the workflow shown in Figure 10 as an extended version of the state-of-the-art workflow for process characterization (left column) that incorporates hDoE (right column, green boxes)and what they represented  aligns with both the FDA and EMA guidelines for process characterization, as the former recommends DoE to increase process knowledge and the latter impurity spiking challenges for downstream operations. hDoE employed in the context of this extended workflow integrates both types of experiment and provides a systematic method for when and where to perform them.

As I said already, there are some empty lines between some sections that are not necessary and the references ar enot accoridng to the journals requirements. Some of the references are old, but authors are citing original statistical methods used in this study, and the other references (there are 14) are newer and up to date.

Author Response

Thank you for the review and your suggestions!

1.) in the discussion part I can see now that in line 400 there is not a link on the section but writes error, so this part should be changed.

The link was removed.

2.) As I said already, there are some empty lines between some sections that are not necessary and the references ar enot accoridng to the journals requirements. Some of the references are old, but authors are citing original statistical methods used in this study, and the other references (there are 14) are newer and up to date.

We tried to align our spacing with the MDPI Bioengineering template and double-checked our text (we only found one paragraph where there were too many line breaks).

Furthermore, we added several references that also include more recent work.

Reviewer 2 Report

1- Authros mentioned at the end of the abstract that their results show that the hDoE approach leads to a > 50% decrease of experiments even for simple cases and at the same time achieves the main goal of process development, validation, and manufacturing to consistently deliver product quality.

What does it mean by that?

2-In the introduction,

Add the motivation of this manuscript.

Add the contribution of this manuscript.

Add the guide to the rest of the sections in this paper.

3- Except equation 1, all other equations need to be cited if they are not yours.

4-Most of the images were place with not good quality. Improve the quality  of the diagrams, in particular 6, 7, 8 and 9

6-Did you conduct some statistical tests to see if your results are significant.

Author Response

Thank you for your review and notes! We try to resolve them point per point:

1.) Figure 6 (a) illustrates the out-of-specification (OOS) rate - which is a direct measure of product quality - over the number of invested runs. One can see that, at around 50 invested total runs (x-axis), the mean OOS rate of the state-of-the-art approach lies at around 8% (dashed line) while hDoE lies around 3% (solid line). This is a +50% decrease in number of total runs for a ground truth process that we consider simple.

2.) Ad motivation: In the introduction we try to illustrate the need for a more flexible process characterization method by highlighting the importance of the PAR in a control strategy and what to do when the PAR is too narrow. Essentially, hDoE is a method for improving the PAR, which is why we chose this as our motivation. This is explicitly stated in line 128 – 131.

Ad contribution: From line 133 we list in three bullet points what our contribution is.

Ad guide: Thank you for this note, as this adds to clarity. We supplemented the corresponding section numbers to the bullet points in line 133, so the reader knows where to find this information.

3.) All other equations represent ground truth equations that we defined ourselves. The point of those is to define a set of relations and coefficients that should be detected by the hDoE procedure.

4.) We included all images in their original, lossless vector data format – which might not be displayed correctly in older version of MS Word. However, if quality refers to the image size, we can ask the editors if there is a way to enlarge them in the manuscript.

5.) Ad significance tests: All models involved in the IPM and the simulation study in general only include significant parameters, yes (with a p-value threshold of 0.05, see table 1). Related to the robustness of results: the simulation studies were repeated 100 times and only statistics like the mean and standard deviation reported and we included the distribution of final parameter estimates in figure 8.

Reviewer 3 Report

The paper explores a new holistic Design of Experiment by using an integrated process model to approach the biopharmaceutical process.

It could be an effective proposal for biopharmaceutical sector.

Author Response

Thank you for the review and your approval of the manuscript!